# Design, development, and validation of multi-epitope proteins for serological diagnosis of Zika virus infections and discrimination from dengue virus seropositivity

Samille Henriques Pereira[1], Mateus Sá Magalhães Serafim[2], Thaís de Fátima Silva Moraes[1], Nathalia Zini[3,4], Jônatas Santos Abrahão[2], Maurício Lacerda Nogueira[3], Jordana Grazziela Alves Coelho dos Reis[1], Flávia Fonseca Bagno[4], Flávio Guimarães da Fonseca[1,4] *

1 Laboratório de Virologia Básica e Aplicada, Departamento de Microbiologia, Universidade Federal de Minas Gerais, Belo Horizonte, Minas Gerais, Brazil, 2 Laboratório de Vírus, Departamento de Microbiologia, Universidade Federal de Minas Gerais, Belo Horizonte, Minas Gerais, Brazil, 3 Laboratório de Pesquisa em Virologia, Faculdade de Medicina de São José do Rio Preto, São José do Rio Preto, São Paulo, Brazil, 4 Centro de Tecnologia em Vacinas, Universidade Federal de Minas Gerais, Belo Horizonte, Minas Gerais, Brazil

* fdafonseca@icb.ufmg.br

**Data Availability Statement:** All data included in the manuscript is freely available by accessing

## Abstract

Zika virus (ZIKV), an arbovirus from the *Flaviviridae* family, is the causative agent of Zika fever, a mild and frequent oligosymptomatic disease in humans. Nonetheless, on rare occasions, ZIKV infection can be associated with Guillain-Barré Syndrome (GBS), and severe congenital complications, such as microcephaly. The oligosymptomatic disease, however, presents symptoms that are quite similar to those observed in infections caused by other frequent co-circulating arboviruses, including dengue virus (DENV). Moreover, the antigenic similarity between ZIKV and DENV, and even with other members of the *Flaviviridae* family, complicates serological testing due to the high cross-reactivity of antibodies. Here, we designed, produced in a prokaryotic expression system, and purified three multiepitope proteins (ZIKV-1, ZIKV-2, and ZIKV-3) for differential diagnosis of Zika. The proteins were evaluated as antigens in ELISA tests for the detection of anti-ZIKV IgG using ZIKV- and DENV-positive human sera. The recombinant proteins were able to bind and detect anti-ZIKV antibodies without cross-reactivity with DENV-positive sera and showed no reactivity with Chikungunya virus (CHIKV)- positive sera. ZIKV-1, ZIKV-2, and ZIKV-3 proteins presented 81.6%, 95%, and 66% sensitivity and 97%, 96%, and 84% specificity, respectively. Our results demonstrate the potential of the designed and expressed antigens in the development of specific diagnostic tests for the detection of IgG antibodies against ZIKV, especially in regions with the circulation of multiple arboviruses.

https://github.com/SamilleHenriques/Paper-ZIKV-multiepitope.

**Funding:** FGF received funds from Fundação de Amparo à Pesquisa do Estado de Minas Gerais - FAPEMIG (grant numbers APQ-03081-17, APQ-04295-17 and RED-00282-16); and from Conselho Nacional de Desenvolvimento Científico e Tecnológico - CNPq (grant # 402256/2022-0). The study was also financed by the Coordenação de Aperfeiçoamento de Pessoal de Nível Superior – Brasil (CAPES) – Finance Code 001, through fellowships given to post-graduation students involved in the work. FGF, MLN, JGACR and JSA are CNPQ-PQ fellowship recipients. The funders had no role in study design, data collection and analysis, decision to publish, or preparation of the manuscript.

**Competing interests:** The authors have declared that no competing interests exist.

## Author summary

From 2015 onward, Zika virus (ZKV) caused major epidemics in Brazil and other South American countries, where many children were born with congenital problems, such as microcephaly, as a result of infections in pregnant women. Additionally, many cases of Guillain Barré Syndrome in adults were also caused by infections with the virus. Detection of ZKV infection is, therefore, very important; nonetheless, most patients infected by ZKV develop either no symptoms or symptoms that are not distinguishable from diseases caused by other co-circulating arboviruses, including dengue. To complicate matters, Zika, and dengue viruses show intense genetic resemblance, and serological diagnosis presents high levels of uncertainty due to extensive cross-reactivity. In this work, we developed proteins to serologically differentiate Zika from dengue infections with no or minimum cross-reactivity. We produced three multiepitope, recombinant proteins which were able to differentiate zika-positive samples from dengue-positive sera. The tests employing the recombinant proteins had high sensitivity and specificity, especially in the case of ZIKV-1 and ZIKV-2 proteins, whilst the ZIKV-3 protein was less efficient. The recombinant proteins showed clear potential to be used as diagnostic tools for detecting antibodies against ZIKV infections with no cross-reactivity to other prevalent arbovirus.

## Introduction

The *Orthoflavivirus zikaense* (ZIKV) is an arthropod-borne virus (arbovirus) transmitted by *Aedes* mosquitoes. It belongs to the *Flaviviridae* family, together with other viruses of public health importance, such as *Orthoflavivirus flavi* (YFV), *Orthoflavivirus denguei* (DENV) and *Orthoflavivirus nilense* (WNV) [1,2]. The emergence of ZIKV in the Americas in 2015 brought great concern, as infections with this virus were associated with serious complications, such as congenital Zika syndrome (CZS). During the epidemics in Brazil, more than 2,630 cases of microcephaly linked to ZIKV infections were registered, in addition to other congenital complications and birth defects, as well as GBS in adults [3]. Nonetheless, most ZIKV infections in humans are asymptomatic, making it difficult to assess the real size of the epidemics. Estimates have put the number of cases in the range of 1.5 million, in Brazil, during the two years of epidemic peak [4]. Although the emergency period ended in July 2017, new cases of CZS and deaths associated with the disease have been reported in low frequencies in the Country. Until the end of 2022, 9,204 probable cases were reported, corresponding to an incidence rate of 4.3 cases per 100,000 Brazilian inhabitants [5,6].

Common clinical manifestations of Zika include fever, rash, headaches, joint pain, conjunctivitis, and muscle pain [7]. The most frequent symptoms are similar to those of other arbovirus diseases, such as dengue and chikungunya, which make clinical diagnosis difficult [8]. The World Health Organization (WHO) recommends the diagnosis by reverse transcriptase quantitative polymerase chain reaction (RT-qPCR), but its use is limited due to the narrow detection window of viral RNA in body fluids [9,10]. Serological tests available on the market, on the other hand, are complicated by high cross-reactivity between different flaviviruses, due to the high genomic and antigenic similarities among them [10]. Cross-reactivity is a problematic issue in places where ZIKV and DENV co-circulate, as is the case in the Americas [11,12]. In this scenario, Plaque Reduction Neutralization Tests (PRNT), which measure specific neutralizing antibodies, could be an alternative; however, the technique is quite laborious and may be imprecise when assessing samples of patients with previous DENV infections, as well as those in the early days after infection [13,14]. As a result, ZIKV diagnosis is not precise, resulting in

under-reporting and misdiagnosing (e.g., DENV). Indeed, our group has previously detected many misdiagnosed or not reported cases of Zika, Dengue, or Chikungunya from suspected patient samples collected during 2015–2017 and sent to the reference state laboratory for diagnosis [15].

The non-structural 1 (NS1) glycoprotein, a protein secreted in large amounts during the viremic phase of flavivirus infections, has been widely used for the development of immunoassays. However, the protein is highly conserved in flaviviruses, presenting up to 62.5% of identity among them [4,16]. Although the NS1 and E proteins are the most immunogenic proteins, all viral proteins have antigenic determinants that could be exploited for precise serological diagnosis [17]. Thus, the use of chimeric proteins with the association of several epitopes that do not show cross-reactivity can increase the performance of immunoassays.

Considering the relevance of ZIKV diagnostics and the need to develop specific and sensitive serological assays for its correct diagnosis, we developed a number of multi-epitope recombinant proteins to be used as antigens and evaluated their specificity and sensitivity in ELISA tests.

## Methods

### Human sera samples and Ethics statement

Sera samples from ten confirmedly positive zika-infected patients (herein termed ZIKV+), ten positive samples for dengue (DENV+), eight positive samples for Chikungunya (CHIKV), and eight samples with elevated levels of rheumatoid factor (RF) were provided by the Laboratory of Virology and Rickettsioses of the Ezequiel Dias Foundation (FUNED). These sera were characterized by commercial ZIKV and DENV IgG kits or RT-PCR, and ultimately characterized by PRNT. Additionally, a second set of sera composed of fifty ZIKV+/DENV- samples, twenty-six DENV+/ZIKV- samples, and fifty-one negative samples for Zika and dengue (ZIKV-/DENV-) were provided by the Virology Research Laboratory of the São José do Rio Preto Medical School (FAMERP). These sera were obtained from a cohort study and characterized by commercial ZIKV and DENV IgG kits, and by PRNT.

Written authorizations to use the sera bank were provided by both Institutions' research boards (Ezequiel Dias Foundation–FUNED, and São José do Rio Preto Medical School—FAMERP). Sera samples from FAMERP were collected upon formal consent from each participant. Sera from FUNED were intended for infection diagnosis and were protected by patient anonymity.

### Multi-epitope recombinant proteins

For the construction of multi-epitope proteins, B cell linear epitopes previously described in the literature and that had been tested in serological assays were annotated [18–21]. From this dataset, epitopes with the lowest percentage of aminoacidic identity between ZIKV and DENV were selected. At the end, seven epitopes were selected, three of them described by Lee et al. (2018) [18], and four epitopes described by Kam et al. (2018) [20]. Three multi-epitope proteins were designed and the epitopes were connected by glycine and proline linkers (GPGPG) as previously described [22]. The amino acid sequences of each final multiepitope polypeptide are shown in **Table 1**. Nucleotide sequences were commercially synthesized and inserted into pET21a+ plasmid (Genone-Biotech, Rio de Janeiro, Brazil) with the addition of the coding sequence for six histidine residues at the COO- end of each protein.

### Modelling and conformational analysis of multi-epitope proteins

Because these were artificially designed proteins, we modeled each one beforehand in order to assess their potential stoichiometric viability. Therefore, the multi-epitope amino acid

**Table 1. FASTA sequences of multi-epitope proteins.** Highlighted are the additional epitopes to each sequence.

| ZIKV-1 | ZIKV-2 | ZIKV-3 |
|---|---|---|
| HKKGEARRSRRAVTLPSHG<br>PGPGTGVFVYNDVEAWRDRYKY<br>GPGPGWGKSYFVRAAKTNNSFVVDGDT<br>LKECPLKHGPGPGPAVIGTAVKGKEAVH | **HMCDATMSYECPMLDEGV**<br>GPGPGHKKGEARRSRRAV<br>TLPSHGPGPGTGVFVYND<br>VEAWRDRYKYGPGPGWGKS<br>YFVRAAKTNNSFVVDGDTLKECP<br>LKHGPGPGPAVIGTAVKGKEAVH | **HMCDATMSYECPMLDEGV**<br>GPGPGHKKGEARRSRRAVT<br>LPSHGPGPG**TVNMAEVRSYC**<br>**YEASIS**GPGPGTGVFVYNDVE<br>AWRDRYKYGPGPG**SVEGELNA**<br>**ILEENGV**GPGPGWGKSYFVRA<br>AKTNNSFVVDGDTLKECPLKH<br>GPGPGPAVIGTAVKGKEAVH |

sequences (FASTA) of ZIKV-1, ZIKV-2, and ZIKV-3 were submitted to the AlphaFold software (version 2.3.1) (https://github.com/sokrypton/ColabFold) [23,24] for prediction of their three-dimensional (3D) protein structures with atomic accuracy. Modeled structures were aligned and analyzed with the PyMOL software (v0.99c) [25]. Figures were also generated with PyMOL.

## Identification of potential binding sites and pockets of multi-epitope proteins

To check whether the potential binding residues and regions were correctly exposed on the AlphaFold2 modeled proteins' surface, we employed the FTSite [26] and FTMap [27] webservers. These modeling tools predict the presence of potential protein cavities that may act as binding sites or hot spots, that is, relevant residues usually responsible for interactions (e.g., protein-protein interactions), such as antigen-antibody ligations. Molecule clusters from the FTMap in each predicted site can also suggest the affinity of these potential binding regions.

## Solvent accessible surface areas (SASA) analyses

The calculation of solvent accessible surface areas (SASA) was assessed with the GETAREA software [28] (https://curie.utmb.edu/getarea.html). Modeled structures were submitted to the webserver in PDB format, resulting in polar, non-polar, and total area/energy of each protein, including the total number of accessible surface atoms and assessment of buried, not accessible atoms. Parameters were set as default for the radius of the water probe (A) = 1.4 with no gradient included in calculations.

## Expression and purification of recombinant proteins

Competent *Escherichia coli* BL21 (DE3) were transformed with the synthesized plasmids and induced for 3 h with 1 mM IPTG (Invitrogen, Massachusetts, USA). Proteins were purified by affinity chromatography using a His-Trap HP column (GE Healthcare, Chicago, Illinois, USA) within the Äkta Start purification system (GE Healthcare Life Sciences, Chicago, Illinois, USA). The proteins were separated by SDS-PAGE and transferred onto nitrocellulose membranes (Cytiva, Danaher Corporation, USA) for western blot evaluation. Subsequently, the membranes were incubated with 1:1000 anti-histidine (Invitrogen, Massachusetts, USA) for 2 h. As a secondary antibody, anti-mouse IgG 1:40,000 (Sigma–Aldrich, San Luis, Missouri, USA) was used. Similarly obtained blots were also incubated with a sera pool from ZIKV + patients (10 sera samples) for 2 h and incubated for another hour with anti-human IgG 1:25,000 (Sigma–Aldrich, San Luis, Missouri, USA). The detection was done using Clarity Western ECL Substrate (Bio-rad, Hercules, California, USA) followed by visualization with image software ChemicDoc (Bio-Rad, Hercules, California, USA).

### In-house indirect ELISA anti-ZIKV (IgG)

The 96-well microplates (High Binding, Sarsterd, Germany, Cat. 82.1581.200) were coated with 400 ng/well of recombinant proteins and incubated overnight at 4˚C. Subsequently, microplates were blocked for 2 h at 37˚C with skim powder milk 5% (Molico–Nestlé, Vevey, Switzerland). Plates were incubated with sera samples ZIKV+, DENV+, and ZIKV-/DENV- (1:100, $Na_2HPO_4$ 100 mM, $KH_2PO_4$ 18 mM, NaCl 1500 mM, KCl 28 mM, sucrose 2%, Tween 20 1%, Tween 80 0,1%, Proclin 0,1%, skim milk powder 5%) for 30 minutes at 37˚C followed by two standard washes, a single wash with urea [6M] solution, and two more standard washes, as previously described [29]. Secondary anti-human IgG antibodies labeled with peroxidase 1:70,000 (Sigma–Aldrich, San Luis, Missouri, USA) were added for 30 minutes at 37˚C. Results were revealed with TMB (Scienco Biotech, Santa Catarina, Brazil). Absorbances were measured at 450 nm in a GO Microplate Spectrophotometer (Thermo Fisher Scientific, Waltham, Massachusetts, USA). The cut-off point was defined by the sum of the mean optical density (OD) of the negative controls plus the standard deviation. The indexes were calculated by the OD of each sample divided by the cut-off value. Results above 1.1 were considered positive; those below 0.8 as negative, and values between 0.8 and 1.1 were classified as borderline results. Positivity is given by the ratio of positive reacting samples over the total of true positive Zika samples. Similarly, sensitivity is given by the ratio of positive reacting samples over true positive samples plus false negative results, removing borderline results from sample totals. Lastly, specificity is given by the ratio of negative reacting samples over the total of true negative samples plus false positive results, also removing borderline results from sample totals.

### Statistical analyses

Statistical analyses were performed using GraphPad Prism 9 program with the Two-way ANOVA test with multiple comparisons. Non-parametric statistics provide an alternative approach to significance testing which could be explored in future research. A receiver operating characteristic (ROC) curve, was applied to evaluate the ELISA test performance as a statistical tool for the global accuracy of the test.

## Results

### Multi-epitope protein modeling, conformational and potential binding sites' analyses, and accessible surface calculations of ZIKV-1, ZIKV-2 e ZIKV-3

Initially, ZIKV multi-epitope protein models were obtained with AlphaFold, and the predicted structures were compared in a 3D alignment, showing their structural differences. ZIKV-1 (Fig 1A) presents a shorter amino acid chain when compared to ZIKV-2 (Fig 1B), which presents a **HMCDATMSYECPMLDEGVGPGPG** polypeptide addition (position 1 to 23). Lastly, both are shorter than ZIKV-3 (Fig 1C), which also presents a **HMCDATMSYECPML-DEGVGPGPG** addition (position 1 to 23), as well as two intercalations **GPGPGTVNMAEVRSYCYEASIS** (position 42 to 63) and **GPGPGSVEGELNAILEENGV** (positions 87 to 106). Interestingly, their 3D alignment (Fig 1D), showed similar spatial occupancy in all three structures, despite their differences in size, but also highlighted their conformational and folding variations, implied from the sequences' additions for ZIKV-2 and ZIKV-3, and both sequences' intercalations within ZIKV-3.

Subsequently, to compare the presence of their respective, potential binding sites, predictions employing FTSite and FTMap softwares were analyzed. First, the three predicted binding sites within the ZIKV-1 structure were shown to be in tight, close formation, according to the

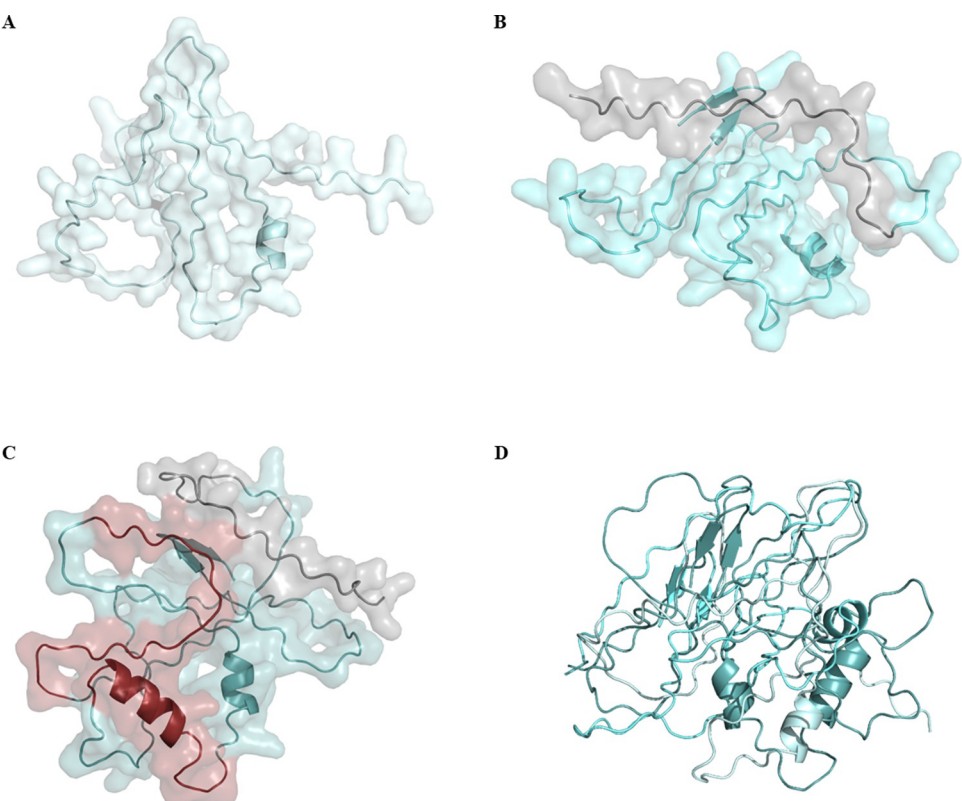

**Fig 1. Models of the multi-epitope proteins.** Modeled structures obtained with AlphaFold are represented by cartoons with transparent surfaces; (A) ZIKV-1 (pale cyan). (B) ZIKV-2 (aquamarine) presents a sequence addition (gray), as well as (C) ZIKV-3 (light teal), which also presents two sequences intercalation (ruby), evidencing structures' differences in size. (D) 3D alignment corroborates their structural differences, also showing a similar spatial occupation among the three proteins and their folding variations. Images were generated with PyMOL (v0.99c).

FTSite (Fig 2A). This was slightly different from the structural prediction in ZIKV-2, showing a slight move of site #1 (salmon) and a shift of site #2 (lime green) towards the added sequence in position 1 to 23 (Fig 2B). Interestingly, ZIKV-3 potential sites were closely related to ZIKV-1, but with a deviation towards the two sequences intercalations (Fig 2C), with both sites being mostly covered by this part of the structure, which could result in a lower sensitivity. In addition, the FTSite results were also corroborated by the probes' clustering predicted with the FTMap (Fig 5D–5F), corroborating their preferable regions in which bindings would occur, such as protein-protein interactions.

Sequence additions (ZIKV-2 and ZIKV-3) and intercalations (ZIKV-3) resulted in conformational and folding changes in the two last predicted structures. Especially for ZIKV-3, which has the longer amino acid sequence, suggesting structural changes or additional protein-protein interactions. To evaluate these potential differences in proteins' accessibility, we also calculated the solvent accessible surface area (SASA) of the three multi-epitope proteins, which showed an evident increase in the number of available surface atoms and total area/energy of interaction for ZIKV-3 (Table 2). Taken together, these would potentially result in more unspecific protein-protein interactions from the sequence additions and/or intercalations, thus supporting the lower specificity observed in the empiric tests. Lastly, smaller differences in SASA calculations between ZIKV-1 and ZIKV-2 could also support their slight differences as observed in the empiric testing.

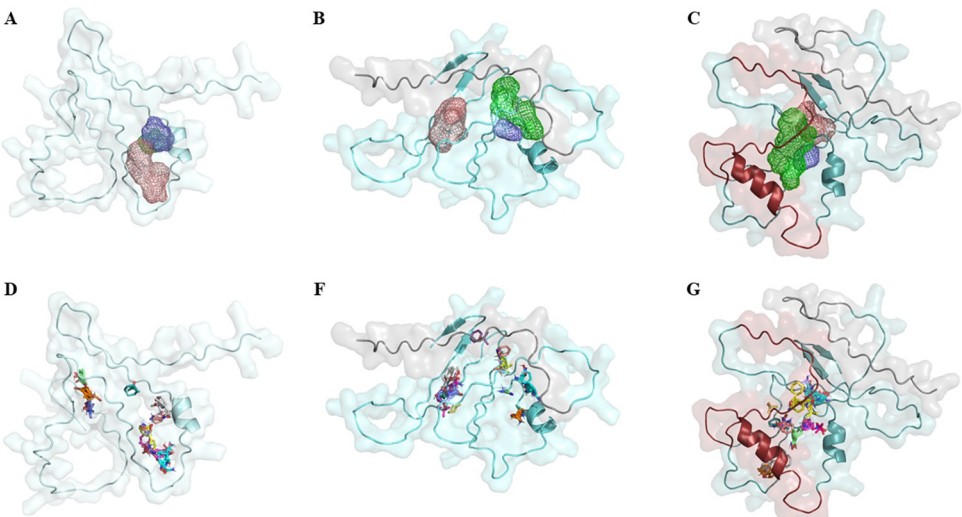

**Fig 2. Potential binding sites and probes clustering predicted in the multi-epitope proteins.** Three potential binding sites were predicted by FTSite: #1 (salmon), #2 (lime green), and #3 (blue), all represented by a tridimensional mesh in (A) ZIKV-1 (pale cyan), (B) ZIKV-2 and (C) ZIKV-3. Probes clustering predicted by FTMap for (D) ZIKV-1, (E) ZIKV-2, and (F) ZIKV-3, corroborate FTSite predictions. All images were generated using PyMOL software (v0.99c).

## Expression and purification of recombinant proteins

ZIKV multi-epitope proteins were produced in transformed *E. coli* and the production was confirmed by SDS-PAGE. Proteins were purified using nickel affinity chromatography (HisTrap) and western blot analysis with a pooled sera sample from ZIKV+ patients (Fig 3).

## Development of an in-house anti-ZIKV IgG ELISA and evaluation of the recombinant multi-epitope proteins as antigens

The multi-epitope proteins produced were evaluated as antigens in an indirect ELISA assay. For the tests, 60 ZIKV+ sera samples; 51 double negative samples (negative for ZIKV and DENV); 36 DENV+ samples; eight CHIKV+ samples; and eight RF+ samples were used. The ZIKV-1 protein (Fig 4A) reacted with 52 of the 60 ZIKV+ samples tested (86.6% positivity), with three samples classified as indeterminate results (91.2% sensitivity); and one out of the 51 negative samples showed a false-positive result (98.1% specificity). In addition, regarding DENV+ samples, only three out of 36 samples showed some reaction with ZIKV-1, and five were placed in the grey zone (90.3% specificity). Lastly, in relation to the eight CHIKV+ and RF+ samples, ZIKV-1 presented no reaction, suggesting 100% specificity. The ZIKV-2 protein (Fig 4B) reacted with 57 out of the 60 ZIKV+ samples with no borderline results (95% positivity and sensitivity), and two negative samples showed false-positive results, also with no indeterminate results (96.2% specificity). Regarding cross-reactivity, it presented 100% specificity against 36 DENV+ samples, with one in the grey zone, and 100% specificity against the CHIKV- and RF-positive sera samples. Lastly, the ZIKV-3 (Fig 4C) protein showed the lowest performance in comparison to the first two multi-epitope proteins, reacting with 32 out of the 60 ZIKV+ sera samples (53.3% positivity), and 12 samples in the grey zone (66.6% sensitivity). In addition, it also showed lower specificity (91%) regarding DENV+ samples, with nine out of 36 samples tested being in the grey zone. As for the CHIKV+ and RF+ samples, tested as possible interferents, ZIKV-3 presented 100% specificity, although one CHIKV+ sera was placed in the grey zone after testing.

**Table 2. SASA calculations for multi-epitope proteins.**

| Parameters | ZIKV-1 | ZIKV-2 | ZIKV-3 |
|---|---|---|---|
| Polar area/energy | 3769.77 | 4131.56 | 5207.06 |
| Apolar area/energy | 8151.01 | 8799.85 | 10512.21 |
| Total area/energy | 11920.78 | 12931.41 | 15719.28 |
| Number of buried/ unavailable atoms | 86 | 124 | 168 |
| Number of surface accessible atoms | 643 | 767 | 1016 |

To confirm the obtained results, the developed ELISA was independently tested at the Virology Research Laboratory (at FAMERP, SP, Brazil) accessing a second panel of sera samples. For this independent evaluation, 36 ZIKV+ samples, 96 double negative samples (negative for ZIKV and DENV), and 96 DENV+ samples were used. ZIKV-1 protein (Fig 5A) reacted with 32 out of the 36 ZIKV+ samples (88.9% positivity), with one sample in the grey zone, and one sample as a false-negative result (88.9% sensitivity). As for the negative samples, three out of the 96 tested presented false-positive results and three were in the grey zone (96.9% specificity). Regarding DENV+ samples as interferents, four samples out of the 96 tested reacted positively (95% specificity). The ZIKV-2 protein (Fig 5B) reacted with 35 out of the 36 samples tested, with no borderline results (97.2% positivity and sensitivity). Among the 96 negative samples tested, three showed false-positive results (96.9% specificity). Regarding DENV+ samples as interferents, ZIKV-2 reacted with only one sample, and one was in the grey zone (98.9% specificity). Similar to the first set of samples, the ZIKV-3 protein (Fig 5C) showed the lowest performance, reacting with 19 out of 36 ZIKV+ samples (52.8% positivity), with 11 in the grey zone (76% sensitivity). Of the 96 negative samples tested, it reacted with one sample, and 17 were in the grey zone (82% specificity). Regarding the 96 DENV+ samples, four showed a false-positive result (94.9% specificity), but a considerable number of samples were in the grey zone (n = 17).

## Discussion

The diagnosis of ZIKV is challenging, as most infected humans do not present readily discernible clinical signs [30]. Furthermore, clinical diagnosis in endemic areas is difficult due to symptoms similarities with other arboviral diseases [31]. To further complicate things, high genetic and antigenic similarities between circulating flaviviruses, especially ZIKV and DENV [11,32] make serological diagnosis imprecise. Currently, Brazil is facing the simultaneous circulation of Dengue, Zika and Chikungunya viruses, with a severe epidemic of Dengue leading to a high number of cases and deaths. Probable cases of Zika are currently low, considering the historical series; however, as of February 2024, 13 probable cases in pregnant women have been reported, two of which have been confirmed [33], showing the need for effective tests in the public health system. Nonetheless, it is worth noting that the system may be overloaded due to the dengue outbreak and therefore it is possible that not all samples are being tested for all arboviruses, and Zika cases may be underreported. Therefore, the search for antigens that allow specific detection of antibodies directed to these viruses is challenging [11,34,35]. Although ZIKV infection is self-limiting, it is important to follow up and monitor cases due to their association with GBS in adults, sequelae in pregnant women (e.g., GBS complications), neurological newborn disorders, congenital syndrome, birth defects, and even perinatal death or stillbirth [35–38].

Regarding diagnosis, full-length, recombinant NS1 is commonly used in commercial anti-ZIKV and anti-DENV ELISA kits. Diagnostic tests also use the E protein, either full-length or

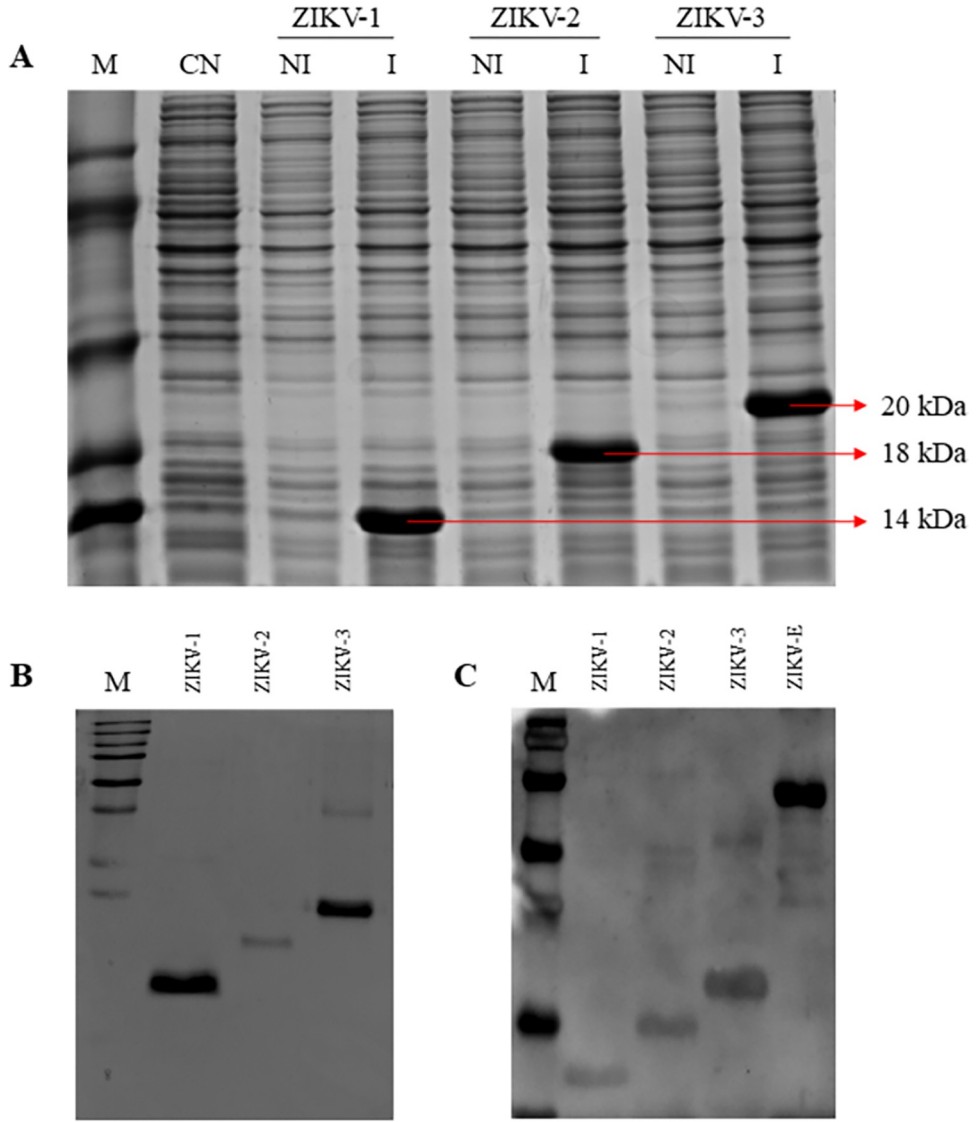

**Fig 3. Production of ZIKV-1, ZIKV-2, and ZIKV-3 multi-epitope proteins.** (A) SDS-PAGE of cell extracts after induction with 1mM IPTG showing the expression of proteins ZIKV-1 (14 kDa), ZIKV-2 (18 kDa), and ZIKV-3 (20 kDa). (B) Multi-epitope proteins after purification by nickel affinity chromatography using HisTrap columns. (C) Western blot of multi-epitope proteins with pooled sera from ZIKV+ patients. M: molecular weight standards; CN: non-transformed cells; NI: transformed and non-induced cells; I: cells induced with 1M IPTG; ZIKV-E: ZIKV envelope protein (recombinant, positive control).

a truncated form in which the transmembrane region is removed to facilitate the recombinant protein production [39]. ZIKV and DENV have about 60% amino acid sequence identity, on average, with E and NS1 proteins having 61.4% and 62.5% identity, respectively [17]. The high homology of these proteins impacts the specificity of available tests, challenging the correct diagnosis of Zika and dengue, especially in endemic areas [40]. Therefore, the development of serological tests based on chimeric recombinant proteins capable of differentiating Zika-positive samples from dengue-positive sera is a feasible option to help with differential diagnosis, even in highly endemic areas [41]. Multi-epitope proteins produced in eukaryotic systems of protein expression usually show satisfactory yields [42], and are satisfactorily antigenic.

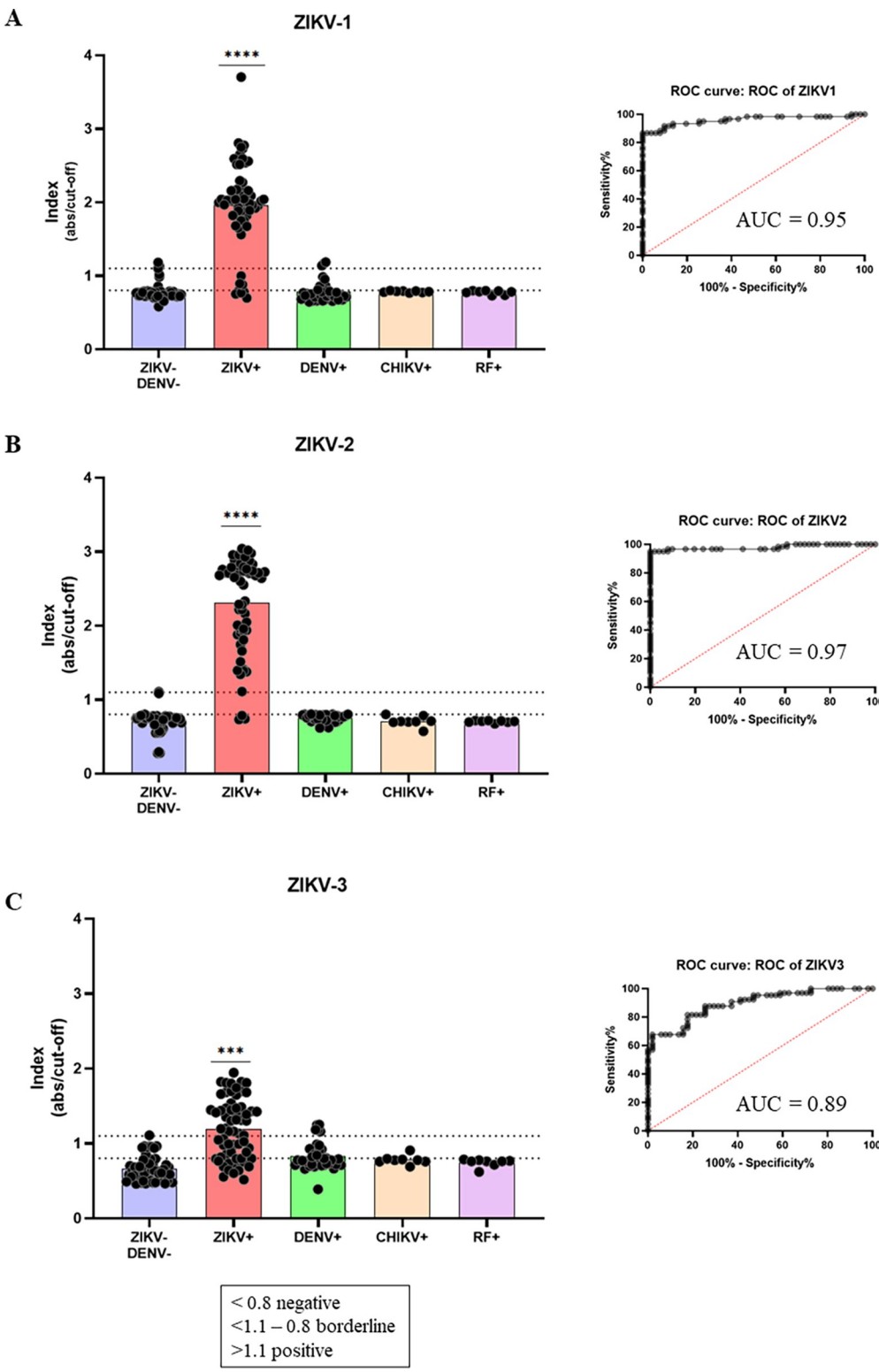

**Fig 4. Indirect anti-ZIKV IgG in-house ELISA.** Multi-epitope proteins were evaluated as antigens in an ELISA, using ZIKV+, DENV+, CHIKV+, RF+ patient samples, and negative samples. (A) ZIKV-1 multi-epitope protein as antigen aside from its corresponding ROC curve. (B) ZIKV-2 multi-epitope protein as antigen and its ROC curve. (C) ZIKV-3 multi-epitope protein as antigen and its ROC curve. Indexes consider the ratio between absorbance readings and cut-off values. Samples dispersions are represented, and significant differences are highlighted with (*** for p>0.001 or

**** for p>0.0001). ZIKV-/DENV-: negative samples for Zika and dengue; ZIKV+: positive samples for Zika; DENV+: positive samples for dengue; CHIKV+: positive samples for CHIKV; RF+: samples with increased levels of rheumatoid factor.

Bacteria-made antigens, are much cheaper to be produced, but may have antigenic problems. Considering the advantages of large-scale production in prokaryotes, we sought to design, produce, and evaluate recombinant proteins that would be promising antigens in a Zika/Dengue differential test.

In the ELISA assays, the multi-epitope proteins showed a satisfactory performance with the first set of sera samples, mainly the ZIKV-1 and ZIKV-2 proteins, where 91.2% and 95% sensitivity were obtained, respectively. In addition, high specificity regarding negative samples was obtained (98.1% and 96.2%, respectively), as well as specificity against DENV+ samples (90.3% and 100%, respectively). In addition, as expected, there was no cross-reactivity with CHIKV + nor with sera from patients with increased FR levels, which is usually considered as a potent interferent in serological tests. Performance assessed with the second set of sera samples corroborated initial results with a similar performance for ZIKV-1 and ZIKV-2 proteins, presenting sensitivity of 88.9% and 97.2%, respectively. In addition, similar results were also observed for their specificity, with 96.9% for both proteins regarding negative samples, as well as 96% and 98.9% specificity regarding DENV+ samples, respectively. The number of samples used in our work was severely limited by sera availability. Due to the difficulties in correctly diagnosing arboviruses during epidemics in affected countries, the number of reliable samples we had access to is not so large. However, at this proof-of-principle stage, we believe the sample panel was sufficient to perform the tests and validations. Further validations of the proposed tests will surely require a larger sample size. The area under the curve (AUC) was 0.97 and 0.95 for ZIKV-1, whilst 0.99 and 0.97 for ZIKV-2, demonstrating a high accuracy, since the AUC represents the overall performance of each antigen, taking into account the sensitivity values of each and their cut-off points. The greater the capability of a test to discriminate between seropositive and seronegative samples, the closer the curve approaches the upper left corner. Thus, the closer the AUC is to 1, the better the test performance [43].

On the other hand, the ZIKV-3 protein showed an overall lower performance regarding sensitivity and specificity on ELISA. One could argue about its larger 3D structure, showing conformational differences and a higher accessible surface, potentially resulting in unspecific protein-protein interactions with other antibodies, such as those from DENV+ samples. This is supported by the binding sites availability and SASA calculations, which could suggest the deviation of antibody interactions to more unspecific fractions of the target protein. This is important as SASA is a decisive factor in protein folding, conformation, and stability studies, including biophysical and thermodynamic properties of the proteins, which would reflect on protein-protein interactions, including the atomic structure and recognition sites of antibody-antigen. Additionally, finding the most bound-like conformation of unliganded protein structures, such as those assessed here, could be related to the presence of both sequences (GPGPGTVNMAEVRSYCYEASIS and GPGPGSVEGELNAILEENGV) intercalations within the ZIKV-3 structure, exposed in its 3D conformation. These could also be associated with unspecific interactions to other antibodies (lower specificity), as aforementioned from the diversified samples used in the tests, as well as hiding (i.e., burying) important residues for specific interactions with ZIKV+ (lower sensitivity). However, no empiric analyses were performed after protein expression, so it cannot be guaranteed that protein folding occurred as predicted after purification.

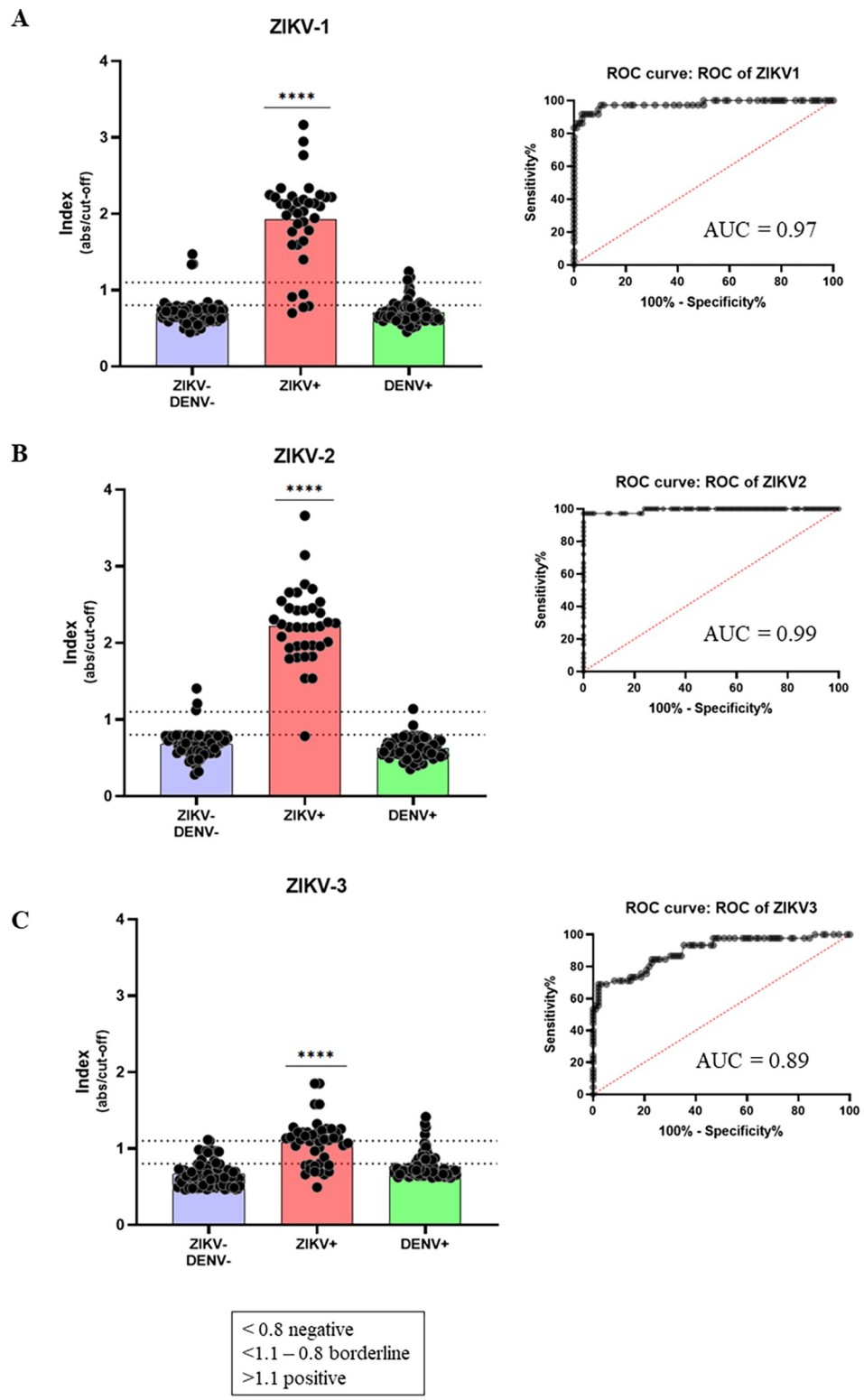

**Fig 5. Indirect anti-ZIKV IgG in-house ELISA, tested at the FAMERP facility, SP, Brazil.** (A) ZIKV-1 multi-epitope protein as antigen and its respective ROC curve. (B) ZIKV-2 multi-epitope protein as antigen and its respective ROC curve. (C) ZIKV-3 multi-epitope protein as antigen and its respective ROC curve. Indexes consider the ratio between absorbance readings and cut-off values. Samples dispersions are represented, and significant differences are highlighted with (**** for $p > 0.0001$). ZIKV-/DENV-: negative samples for Zika and dengue; ZIKV+: positive samples for Zika; DENV+: positive samples for dengue.

In our study, the accessed sera samples came from two different states: from several cities in Minas Gerais state, Brazil, obtained during the Zika outbreak in 2015; and from São Paulo state, Brazil, from a cohort study in the city of São José do Rio Preto. This is important because we could validate the antigenicity of the multiepitope constructs in different epidemiological settings, providing a diversity of samples. Furthermore, the epitopes present in the proteins show high conservation between strains of African and Asian origin. An analysis made with ten strains of African origin and thirty-five strains of Asiatic origin shows that the Epitope 1 (HMCDATMSYECPMLDEGV) has 100% similarity with African strains and 100% with Asiatic strains. The Epitope 2 (HKKGEARRSRRAVTLPSH) has 100% similarity with African strains and only one of the Asian strains showed a difference in one amino acid when compared to the epitope sequence. The Epitope 3 (TVNMAEVRSYCYEASIS) has 88% similarity with the African and Asian strains. The Epitope 4 (TGVFVYNDVEAWRDRYKY) has a valine amino acid in position 5, five strains also have the valine amino acid while five other strains have the isoleucine amino acid, however, they have 100% similarity with the Asiatic strains. The Epitope 5 (SVEGELNAILEENGV) has 100% similarity with African strains and 100% similarity with thirty-one Asian strains and one amino acid difference with the other four strains. The Epitope 6 (WGKSYFVRAAKTNNSFVVDGDTLKECPLKH) has 100% similarity with nine African strains and presented a difference of one amino acid for one strain and 100% similarity with thirty-three Asian strains and presented a difference of one amino acid for two strains. The Epitope 7 (PAVIGTAVKGKEAVH) has 100% similarity with nine African strains and presented a difference of one amino acid for one strain and 100% similarity with thirty-two Asian strains and a difference of one amino acid with three strains (Supplementary material). This high conservation suggests that the multiepitope proteins can be used not only with Brazilian samples but also with samples from other countries.

The developed tests presented superior results compared to available commercial kits, since it was possible to maintain high sensitivity and specificity and, importantly, low cross-reactivity with DENV+ sera. For instance, Low et al. (2021), evaluated two Zika IgG commercial ELISA tests, both of which use the NS1 protein as an antigen, using a panel comprising 278 sera or plasma samples [44]. The Diapro ZIKV IgG test showed an overall specificity of 76.7% and a sensitivity of 84.2%, and when evaluated against DENV+ patients, it showed 8.3% positivity for primary infections and 55.6% for secondary infections. The Euroimmun ZIKV IgG test on the other hand showed an overall specificity of 90%, but a low sensitivity of 52.6%. In addition, when evaluating the test with DENV+ patients, authors found no positivity for primary infections and 22.2% for secondary infections [44]. Morales and coworkers also evaluated the Euroimmun ZIKV IgG test with 907 samples and observed a low sensitivity of 41.4% for acute phase samples, and 72% sensitivity for convalescent samples. As for the negative samples, 9.4% of positivity was obtained, while 33.3% of DENV+ samples reacted positively [45]. Similarly, L'Huillier et al. (2017) also evaluated the Euroimmun test with 223 samples and obtained an overall sensitivity of 23.7% and a specificity of 95.2% [46].

In this work, we highlight the relevance of the use of chimeric proteins as an alternative and potential candidate for differential diagnosis of Zika, especially when considering dengue-positive samples. Additionally, the combination of epitopes derived from several distinct viral proteins with low reactivity among flaviviruses is clearly advantageous when compared to the use of full-length proteins. The successful use of multi-epitope chimeric proteins has been increasingly documented in the literature, and includes chimeric proteins for the diagnosis of *Trypanosoma cruzi* [47], *Burkholderia pseudomallei* [48], *Human T-cell lymphotropic virus* [49,50], *Brucella* [51], *Leishmania braziliensis* [52], among others. These publications also demonstrate the reliability and feasibility of a multi-epitope recombinant approach for diagnosis. Further

developments of the diagnostic tool proposed here will include testing IgM positive samples for Dengue or Zika, as IgM is an important marker for either recent or ongoing infection.

## Supporting information

**S1 Table. Linear epitopes for B cells found in the literature and tested in serological tests for Zika virus.**
(DOCX)

**S1 Fig. Alignment of African and Asian strains with the epitopes that compose the multie-pitope proteins.** (A) Alignment of Epitope 1 with ten African strains and thirty-five Asian strains. (B) Alignment of Epitope 2 with ten African strains and thirty-five Asian strains. (C) Alignment of Epitope 3 with ten African strains and thirty-five Asian strains. (D) Alignment of Epitope 4 with ten African strains and thirty-five Asian strains. (E) Alignment of Epitope 5 with ten African strains and thirty-five Asian strains. (F) Alignment of Epitope 6 with ten African strains and thirty-five Asian strains. (G) Alignment of Epitope 7 with ten African strains and thirty-five Asian strains.
(PDF)

## Acknowledgments

The authors are very thankful to the Laboratory of Virology and Rickettsioses of the Ezequiel Dias Foundation (FUNED) for providing sera samples. We also thank Edel Figueiredo Barbosa Stancioli for helpful discussions, and Pedro A. Alves for aiding with bioinformatic analyses.

## Author Contributions

**Conceptualization:** Samille Henriques Pereira, Flávio Guimarães da Fonseca.

**Data curation:** Flávio Guimarães da Fonseca.

**Formal analysis:** Samille Henriques Pereira, Mateus Sá Magalhães Serafim, Thaís de Fátima Silva Moraes, Nathalia Zini, Jônatas Santos Abrahão, Maurício Lacerda Nogueira, Jordana Grazziela Alves Coelho dos Reis, Flávia Fonseca Bagno, Flávio Guimarães da Fonseca.

**Funding acquisition:** Flávio Guimarães da Fonseca.

**Investigation:** Samille Henriques Pereira.

**Methodology:** Samille Henriques Pereira, Thaís de Fátima Silva Moraes, Nathalia Zini, Flávia Fonseca Bagno.

**Project administration:** Flávio Guimarães da Fonseca.

**Software:** Mateus Sá Magalhães Serafim.

**Supervision:** Flávio Guimarães da Fonseca.

**Validation:** Nathalia Zini, Flávia Fonseca Bagno.

**Writing – original draft:** Samille Henriques Pereira, Flávio Guimarães da Fonseca.

**Writing – review & editing:** Samille Henriques Pereira, Mateus Sá Magalhães Serafim, Thaís de Fátima Silva Moraes, Nathalia Zini, Jônatas Santos Abrahão, Maurício Lacerda Nogueira, Jordana Grazziela Alves Coelho dos Reis, Flávia Fonseca Bagno, Flávio Guimarães da Fonseca.

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
