## [Decision Letter · Decision Letter 0]

13 Feb 2024

Dear Dr Da Fonseca,

Thank you very much for submitting your manuscript "Design, development, and validation of multi-epitope proteins for serological diagnosis of Zika virus infections and discrimination from dengue virus seropositivity" for consideration at PLOS Neglected Tropical Diseases. As with all papers reviewed by the journal, your manuscript was reviewed by members of the editorial board and by several independent reviewers. The reviewers appreciated the attention to an important topic. Based on the reviews, we are likely to accept this manuscript for publication, providing that you modify the manuscript according to the review recommendations. 

Sincerely,

Michael W Gaunt, PhD

Academic Editor

Andrea Marzi

Section Editor

AE comments

This is an impressive manuscript, very well written and well designed. There are two minor issues,

line 151-152

"For the construction of multi-epitope, recombinant proteins, known epitopes described in the literature and tested in serological assays were annotated."

Can the authors rephrase this sentence because it isn't clear and an appropriate reference would be useful.

line 233-234

"Statistical analyses were performed using GraphPad Prism 9 program with the Two-way ANOVA test with multiple comparisons. "

Its "two-way ANOVA" (small "t"). To avoid controversy - the authors have not checked whether the conditions of ANOVA are met, kindly add the following statement,

"Non-parametric statistics provide an alternative approach to significance testing which could be explored in future research."

This is to provide cover for the authors should the Fmax test fail by a future investigator. If the authors have a strong Fmax value the above sentence can be ignored - providing the value is reported. Generally the key conditions for ANOVA have been met, i.e. they are truly independent and "homogeneity of variance" in the authors data looks about right. The ROC is the right test here so thats fine. 

Reviewers comments

A huge thanks to all reviewers for their time and expertise.

Reviewer 1 is requesting sera of "African or Asian origin". This will be extremely difficult for the authors to obtain. One issue is Brazil has very strict biomaterial exportation laws and this makes reciprocation difficult, also the scarcity of this sera will make it hard. I agree bioinformatics analyse will address Reviewer 1's request, i.e. amino acid homology and predicted antigenicity. All other requests by the Reviewers are helpful for strengthening the manuscript and should be addressed.

Please note one potential reviewer (not represented here), accepted the invitation to review, but did not submit a report and led to delays in the turn around time.

Reviewer's Responses to Questions

**Key Review Criteria Required for Acceptance?**

**Methods**

-Are the objectives of the study clearly articulated with a clear testable hypothesis stated?

-Is the study design appropriate to address the stated objectives?

-Is the population clearly described and appropriate for the hypothesis being tested?

-Is the sample size sufficient to ensure adequate power to address the hypothesis being tested?

-Were correct statistical analysis used to support conclusions?

-Are there concerns about ethical or regulatory requirements being met?

Reviewer #1: see below

Reviewer #2: -Are the objectives of the study clearly articulated with a clear testable hypothesis stated? YES

-Is the study design appropriate to address the stated objectives? YES

-Is the population clearly described and appropriate for the hypothesis being tested? YES 

-Is the sample size sufficient to ensure adequate power to address the hypothesis being tested? I CANNOT OPINION

-Were correct statistical analysis used to support conclusions? I CANNOT OPINION

-Are there concerns about ethical or regulatory requirements being met? YES

Reviewer #3: Overall, the study demonstrates a well-defined methodology, population description, and statistical analysis to address the stated objectives. However, explicit justification of sample size would help the manuscript. There are some concerns about the number of Chikungunya and RF+ samples used, but this is more in the context of next steps for the test developed not the manuscript. This is why a justification of the sample size at this stage is critical not necessarily an increase in sample size.

**Results**

-Does the analysis presented match the analysis plan?

-Are the results clearly and completely presented?

-Are the figures (Tables, Images) of sufficient quality for clarity?

Reviewer #1: see below

Reviewer #2: -Does the analysis presented match the analysis plan? YES

-Are the results clearly and completely presented? YES

-Are the figures (Tables, Images) of sufficient quality for clarity? YES

Reviewer #3: The results are presented in a structured manner, with clear subsections describing different aspects of the analysis (e.g., protein modeling, purification, antigen evaluation). Figures and tables are clear and are easy to read and understand. Visually they support the findings. Quantitative data, including sensitivity, specificity, and ROC curve analyses, are reported to quantify the performance of the diagnostic assays.

**Conclusions**

-Are the conclusions supported by the data presented?

-Are the limitations of analysis clearly described?

-Do the authors discuss how these data can be helpful to advance our understanding of the topic under study?

-Is public health relevance addressed?

Reviewer #1: see below

Reviewer #2: -Are the conclusions supported by the data presented? YES

-Are the limitations of analysis clearly described? YES

-Do the authors discuss how these data can be helpful to advance our understanding of the topic under study? YES

-Is public health relevance addressed? YES

Reviewer #3: The conclusions are supported by the data presented. For the most part the limitations of analysis are clearly designed. Any concerns on this point are highlighted in the summary.

Public health relevance could be improved slightly by including a sentence on the level of ZIKV transmission currently.

**Editorial and Data Presentation Modifications?**

Reviewer #1: see below

Reviewer #2: (No Response)

Reviewer #3: Minor suggestions are included in the summary and general comments section.

**Summary and General Comments**

Reviewer #1: This study describes the design and expression of recombinant proteins comprised of antibody epitopes specific for Zika virus, and predicted to not cross react with anti-dengue virus sera. Distinguishing between ZIKV and DENV (or other flaviviruses) in infected individuals is potentially quite significant, as it may influence subsequent medical interventions. The epitopes are within the NS1 protein, and have previously been described. Three recombinant proteins were expressed, containing 4 (ZIKV-1), 5 (ZIKV-2) or 6 NS1 epitopes divided by spacers of 5-amino acids. The proteins were used to coat ELISA plates and tested against sera from to cohorts of patients with confirmed ZIKV, DENV, or CHIKV infection. The smaller two recombinant proteins (ZIKV-1 and ZIKV-2) showed sensitivity and specificity that exceeded reported values for several commercially available tests. 

In general, this is a well written manuscript that clearly shows the utility of the bacterially expressed multi-epitope proteins in distinguishing ZIKV infection from DENV, and likely other flavivirus infections. However, clarification of a few points would strengthen the overall impact.

1. The cohorts used to test the sensitivity/ specificity are both of Brazilian origin, indicating infection with an Asian-American lineage ZIKV. Do the ZIKIV epitope recombinant proteins (or ZIKV-2, specifically) interact with antibodies from sera of African or Asian origin? If these cohorts are not available, could the authors can provide information about the conservation of the chosen epitopes in Asian or African ZIKV lineages?

2. Can the recombinant proteins be used to detect specific IgM responses? Because identification of the infecting virus may be of the most benefit as early in infection as possible, assessing the IgM response may increase the significance of this assay.

Reviewer #2: (No Response)

Reviewer #3: The study investigated the use of multi-epitope ZIKV proteins, where each epitope set 1, 2 and 3 include an increasing number of epitopes. The manuscript approach then utilizes AlphaFold for modeling and predicting structural variations. Comparative analysis revealed distinct structural differences, including sequence additions and intercalations, impacting potential binding sites and protein accessibility. Following protein expression and purification, antigenicity was evaluated using ELISA assays with ZIKV-positive, dengue virus (DENV)-positive, and negative sera samples. While ZIKV-1 and ZIKV-2 exhibited robust sensitivity and specificity in detecting ZIKV antibodies, ZIKV-3 showed lower performance, potentially attributed to its structural modifications. The results suggest the potential utility of certain multi-epitope proteins as effective antigens for ZIKV detection, emphasizing the importance of structural considerations in antigen design and evaluation for diagnostic purposes.

There are several minor concerns/weaknesses with the current manuscript. Overall, the study presents a promising approach to address the diagnostic challenges associated with ZIKV infection. At this point in the development of this test, it is best to address the following questions within the text. It may help further validation and consideration of potential limitations to ensure the reliability and clinical relevance of the developed diagnostic test in future work.

Minor concerns:

The cut-off’s for the ELISA’s as described in lines 221-226, What samples were used to determine the negative? Was it sample set 1 (FUNED) or sample set 2 (FAMERP)? Which protein was used ZIKV 1, 2 or 3?

What is the rationale for the use of Urea in one of the ELISA washes? Were pilot experiments completed without the Urea? And if yes what were the overarching results. In most common CLIA certified ELISA tests Urea is not commonly included in one of the wash steps.

The table of all original epitopes from the literature should be included as supplemental data or an external link included. Are the previous identified individual epitopes known or thought to be linear epitopes or are they conformational epitopes?

The discussion should include future directions for this specific test.

PLOS authors have the option to publish the peer review history of their article (what does this mean?). If published, this will include your full peer review and any attached files.

Reviewer #1: No

Reviewer #2: No

Reviewer #3: Yes: James D Brien

Figure Files:

Data Requirements:

Reproducibility:

References

---

## [Decision Letter · Decision Letter 1]

22 Mar 2024

Dear Dr Da Fonseca,

We are pleased to inform you that your manuscript 'Design, development, and validation of multi-epitope proteins for serological diagnosis of Zika virus infections and discrimination from dengue virus seropositivity' has been provisionally accepted for publication in PLOS Neglected Tropical Diseases.

Best regards,

Michael W Gaunt, PhD

Academic Editor

Andrea Marzi

Section Editor

This is an excellent manuscript universally accepted by all reviewers, on a hugely important area of epidemiology separating ZIKV from DENV infection using a very modern, molecular approach that has produced excellent results.

Reviewer's Responses to Questions

**Key Review Criteria Required for Acceptance?**

**Methods**

-Are the objectives of the study clearly articulated with a clear testable hypothesis stated?

-Is the study design appropriate to address the stated objectives?

-Is the population clearly described and appropriate for the hypothesis being tested?

-Is the sample size sufficient to ensure adequate power to address the hypothesis being tested?

-Were correct statistical analysis used to support conclusions?

-Are there concerns about ethical or regulatory requirements being met?

Reviewer #1: Previous concerns are addressed in the revised manuscript

Reviewer #2: (No Response)

Reviewer #3: (No Response)

**Results**

-Does the analysis presented match the analysis plan?

-Are the results clearly and completely presented?

-Are the figures (Tables, Images) of sufficient quality for clarity?

Reviewer #1: see above

Reviewer #2: (No Response)

Reviewer #3: (No Response)

**Conclusions**

-Are the conclusions supported by the data presented?

-Are the limitations of analysis clearly described?

-Do the authors discuss how these data can be helpful to advance our understanding of the topic under study?

-Is public health relevance addressed?

Reviewer #1: see above

Reviewer #2: (No Response)

Reviewer #3: (No Response)

**Editorial and Data Presentation Modifications?**

Reviewer #1: (No Response)

Reviewer #2: (No Response)

Reviewer #3: (No Response)

**Summary and General Comments**

Reviewer #1: (No Response)

Reviewer #2: (No Response)

Reviewer #3: The authors have provided clear responses to the reviewers questions, leading to an improved manuscript which is rigorous and reproducible.

PLOS authors have the option to publish the peer review history of their article (what does this mean?). If published, this will include your full peer review and any attached files.

Reviewer #1: No

Reviewer #2: No

Reviewer #3: **Yes: **James Brien

---

## [Editor Report · Acceptance letter]

2 Apr 2024

Dear Dr Da Fonseca,

We are delighted to inform you that your manuscript, "Design, development, and validation of multi-epitope proteins for serological diagnosis of Zika virus infections and discrimination from dengue virus seropositivity," has been formally accepted for publication in PLOS Neglected Tropical Diseases.

Best regards,

Shaden Kamhawi

co-Editor-in-Chief

Paul Brindley

co-Editor-in-Chief
